# Mitochondrial fusion supports increased oxidative phosphorylation during cell proliferation

Cong-Hui Yao[1], Rencheng Wang[1], Yahui Wang[1], Che-Pei Kung[2,3], Jason D Weber[2,3], Gary J Patti[3]*

[1]Department of Chemistry, Washington University, St. Louis, United States; [2]Division of Molecular Oncology, Washington University School of Medicine, St. Louis, United States; [3]Department of Medicine, Washington University School of Medicine, St. Louis, United States

**Abstract** Proliferating cells often have increased glucose consumption and lactate excretion relative to the same cells in the quiescent state, a phenomenon known as the Warburg effect. Despite an increase in glycolysis, however, here we show that non-transformed mouse fibroblasts also increase oxidative phosphorylation (OXPHOS) by nearly two-fold and mitochondrial coupling efficiency by ~30% during proliferation. Both increases are supported by mitochondrial fusion. Impairing mitochondrial fusion by knocking down mitofusion-2 (Mfn2) was sufficient to attenuate proliferation, while overexpressing Mfn2 increased proliferation. Interestingly, impairing mitochondrial fusion decreased OXPHOS but did not deplete ATP levels. Instead, inhibition caused cells to transition from excreting aspartate to consuming it. Transforming fibroblasts with the *Ras* oncogene induced mitochondrial biogenesis, which further elevated OXPHOS. Notably, transformed fibroblasts continued to have elongated mitochondria and their proliferation remained sensitive to inhibition of Mfn2. Our results suggest that cell proliferation requires increased OXPHOS as supported by mitochondrial fusion.
DOI: https://doi.org/10.7554/eLife.41351.001

*For correspondence: gjpattij@wustl.edu

## Introduction

Depending on cell type and microenvironment, various adaptations in metabolism have been associated with cellular proliferation. The metabolic adaptation that has received the most attention is a phenomenon known as aerobic glycolysis or the Warburg effect, which is characterized by a high level of glucose consumption and a high rate of glucose fermentation to lactate irrespective of oxygen availability (*Liberti and Locasale, 2016*; *Vander Heiden et al., 2009*). Although the Warburg effect is recognized as a typical feature of dividing cancer cells and is the basis for imaging many tumors in the clinic with fluorodeoxyglucose positron emission tomography (*Hanahan and Weinberg, 2011*; *Zhu et al., 2011*), it is also found in normal proliferating cells such as non-transformed fibroblasts, lymphocytes, macrophages, thymocytes, endothelial cells, and embryonic stem cells (*Brand, 1985*; *Hedeskov, 1968*; *Hume et al., 1978*; *Munyon and Merchant, 1959*; *Wang et al., 1976*). Accordingly, even in non-cancerous contexts, the Warburg effect has been classified as a hallmark of rapid proliferation (*Abdel-Haleem et al., 2017*).

Although there is a general consensus that glycolytic flux increases in proliferating cells, the extent to which oxidative metabolism is altered has been historically complicated (*DeBerardinis et al., 2007*; *Seyfried, 2015*). Warburg originally proposed that cancer cells rely on enhanced glycolysis because of defects in mitochondria (*Warburg, 1956*). Some cancers do have defective mitochondrial enzymes (e.g. succinate dehydrogenase and fumarase), but it is now well

**eLife digest** Most cells in the body contain many small compartments called mitochondria. These tiny powerhouses can use oxygen to break down molecules of glucose (a type of sugar) and release the energy that fuels many life processes. Mitochondria can also use oxygen to build certain compounds essential for the cell.

Rapidly dividing cells, such as the ones found in tumors, need a lot of energy. Yet, they often 'choose' to burn much of their glucose through fermentation, a less efficient process that does not require oxygen or mitochondria. In fact, many theories suggest that cells which divide a lot decrease the quantity of oxygen their mitochondria consume. It is still unclear what role mitochondria have during phases of intense growth, and if they act differently in cancerous and healthy cells.

Here, Yao et al. use a cell system where division can be turned on or off, and find that when cells quickly multiply, their mitochondria actually consume more oxygen. Further experiments then reveal that, in both cancerous and healthy cells, the different mitochondria inside a cell merge during periods of intense division. This mechanism allows the compartment to better use oxygen. Yao et al. go on to show that adjusting the fusion process through genetic manipulation helps to control division. When mitochondria cannot combine, cells divide less well; when the compartments can merge more easily, cells multiply faster.

If growing cells do not rely on their mitochondria for their energy demands during multiplication, why do these compartments seem to be essential for division? The reason might be that the mitochondria produce aspartate, a molecule that cells use to replicate.

The work by Yao et al. suggests that at least certain cancer cells may increase their consumption of oxygen to sustain their mitochondria; armed with this knowledge, it may be possible to design new diagnostic tests and new treatments to identify, and potentially target these oxygen-dependent tumor cells.

DOI: https://doi.org/10.7554/eLife.41351.002

established that most proliferating cells (including cancer) have functional mitochondria (*Ahn and Metallo, 2015*; *Vyas et al., 2016*). Indeed, functional mitochondria are essential to the proliferation of some cell types. Studies have shown that oxidative phosphorylation (OXPHOS) may provide the majority of ATP during proliferation and function to support the synthesis of important molecular building blocks such as aspartate (*Birsoy et al., 2015*; *Fan et al., 2013*; *Rodríguez-Enríquez et al., 2010*; *Sullivan et al., 2015*; *Zu and Guppy, 2004*). Elevated levels of OXPHOS, however, may not necessarily be required to fulfill such functions. To the contrary, many reports have suggested that proliferating cells suppress mitochondrial respiration and statements that glycolysis is preferred over OXPHOS during proliferation are prevalent in the literature (*Whitaker-Menezes et al., 2011*). Certain cancers of the bladder, breast, and kidney are depleted of mitochondrial DNA and have decreased expression of respiratory genes (*Reznik et al., 2016*). Some cancer cells exhibit high levels of mitochondrial fission and have associated decreases in respiratory capacity as mediated by an imbalance of dynamin-related protein 1 (DRP1) and mitofusin-2 (Mfn2) (*Chen and Chan, 2017*; *Rehman et al., 2012*; *Serasinghe et al., 2015*; *Xie et al., 2015*). In other cases, increasing glucose oxidation by inhibiting pyruvate dehydrogenase kinase has been shown to slow the proliferation of transformed cells (*Bonnet et al., 2007*).

A challenge of quantitating changes in OXPHOS as a function of proliferation has been the confounding experimental factors that are often associated with cancer studies. Tumors contain non-proliferating cell types that may shift the average of metabolic measurements from bulk tissues. Additionally, cancer cells from tumors often have restricted access to oxygen (*Brahimi-Horn et al., 2007*). Although oxygen limitation can similarly lead to an enhanced glycolytic phenotype, this metabolic program is distinct from the Warburg effect. Finally, many studies have focused on the proliferative state of cancer cells without having an appropriately matched non-proliferating comparison with the same genetic background tested under the same conditions (*Zu and Guppy, 2004*).

In this study, to directly compare oxidative metabolism in the same cells of the quiescent and proliferative state, we exploited the cell-density-dependent phenotype of non-transformed fibroblasts. We find that even though proliferating fibroblasts exhibit enhanced glycolysis that is consistent with

a Warburg phenotype, they also increase OXPHOS by nearly two-fold and increase their mitochondrial coupling efficiency by ~30%. Interestingly, both increases are supported by mitochondrial fusion. Although transformation with the *Ras* oncogene further elevated OXPHOS, the additional increase was supported by mitochondrial biogenesis rather than changes in mitochondrial dynamics. Blocking mitochondrial fusion slowed proliferation in both non-transformed and transformed cells. Taken together, our results indicate that proliferation of fibroblasts requires an increase in OXPHOS supported by mitochondrial fusion.

## Results

### Proliferation increases oxidative phosphorylation and mitochondrial coupling efficiency

Mouse 3T3-L1 fibroblasts are immortalized, non-transformed cells that retain sensitivity to contact inhibition (*Green and Kehinde, 1975*). They provide a simple, well-controlled model to compare metabolism in the proliferative and quiescent states, as has been demonstrated previously (*Yao et al., 2016a*). The first step in our analysis was to verify that proliferating fibroblasts exhibit the Warburg effect. Relative to quiescent fibroblasts in the contact-inhibited state, proliferating cells had increased glucose consumption and lactate excretion (*Figure 1A*). As expected, proliferating cells excreted a greater percentage of glucose as lactate (47%) compared to quiescent cells (32%) (*Figure 1—source data 1*). Of note, the absolute amount of glucose having a non-lactate fate was also increased by over two-fold in the proliferative state (0.38 pmol/cell/hr) relative to the quiescent state (0.16 pmol/cell/hr) (*Figure 1—source data 1*). Glucose carbon that is not excreted as lactate is potentially available to support an increased rate of oxidative metabolism, which we next aimed to quantify.

Strikingly, on a per cell basis, we found that the basal respiration rate was ~81% higher during proliferation compared to quiescence (*Figure 1—figure supplement 1*). Given that proliferating cells are larger in size relative to quiescent cells, we also independently normalized the oxygen-consumption data by protein content instead of cell number. Even when normalized by protein content, the respiration rate of proliferating cells was ~59% higher than quiescent cells (*Figure 1B–C*). Intriguingly, proliferating cells also exhibited a decrease in proton leak and a ~ 112% increase in ATP production (*Figure 1B–C*). Taken together, the coupling efficiency of proliferating cells was determined to be 34% higher than quiescent cells. We note that the coupling efficiency was calculated as the ratio of the ATP production rate and the basal respiration rate, which is therefore independent of sample normalization method. These data suggest that proliferating fibroblasts with Warburg metabolism not only have increased OXPHOS, but also that they respire more efficiently.

### Glutamine is the major carbon source for fueling the TCA cycle and mitochondrial respiration

We next aimed to investigate which carbon sources fuel mitochondrial respiration by analyzing the utilization of glucose, glutamine, and fatty acids. In addition to increasing glucose consumption and lactate excretion (*Figure 1A*), proliferating fibroblasts take up two-fold more glutamine compared to quiescent fibroblasts (*Figure 1D*). Since the rate of glutamate excretion was unchanged, more glutamine carbon was being used for anaplerosis. We also found that the consumption rates of palmitate and oleate were increased in proliferation by 194% and 98%, respectively (*Figure 1E*).

Since proliferating fibroblasts showed increased utilization of all three nutrients we examined, we next applied stable isotope tracing and metabolomics to access the relative contribution of each carbon source to the TCA cycle. In three separate experiments, cells were fed either uniformly $^{13}$C-labeled glucose (U-$^{13}$C glucose), uniformly $^{13}$C-labeled glutamine (U-$^{13}$C glutamine), or uniformly $^{13}$C-labeled palmitate (U-$^{13}$C palmitate) for 6 hr. The isotope enrichments in citrate, a representative TCA cycle intermediate, were evaluated to infer the relative contribution of each carbon source to the TCA cycle and mitochondrial respiration. Even though proliferating cells consumed more glucose and more fatty acids, citrate labeling from these two carbon sources was significantly decreased in the proliferative state (*Figure 1F*; *Figure 1G*). In contrast, labeling of citrate by glutamine was substantial and significantly increased in proliferating cells relative to quiescent cells, suggesting that glutamine is a major energy source to fuel mitochondrial respiration (*Figure 1H*). This result is

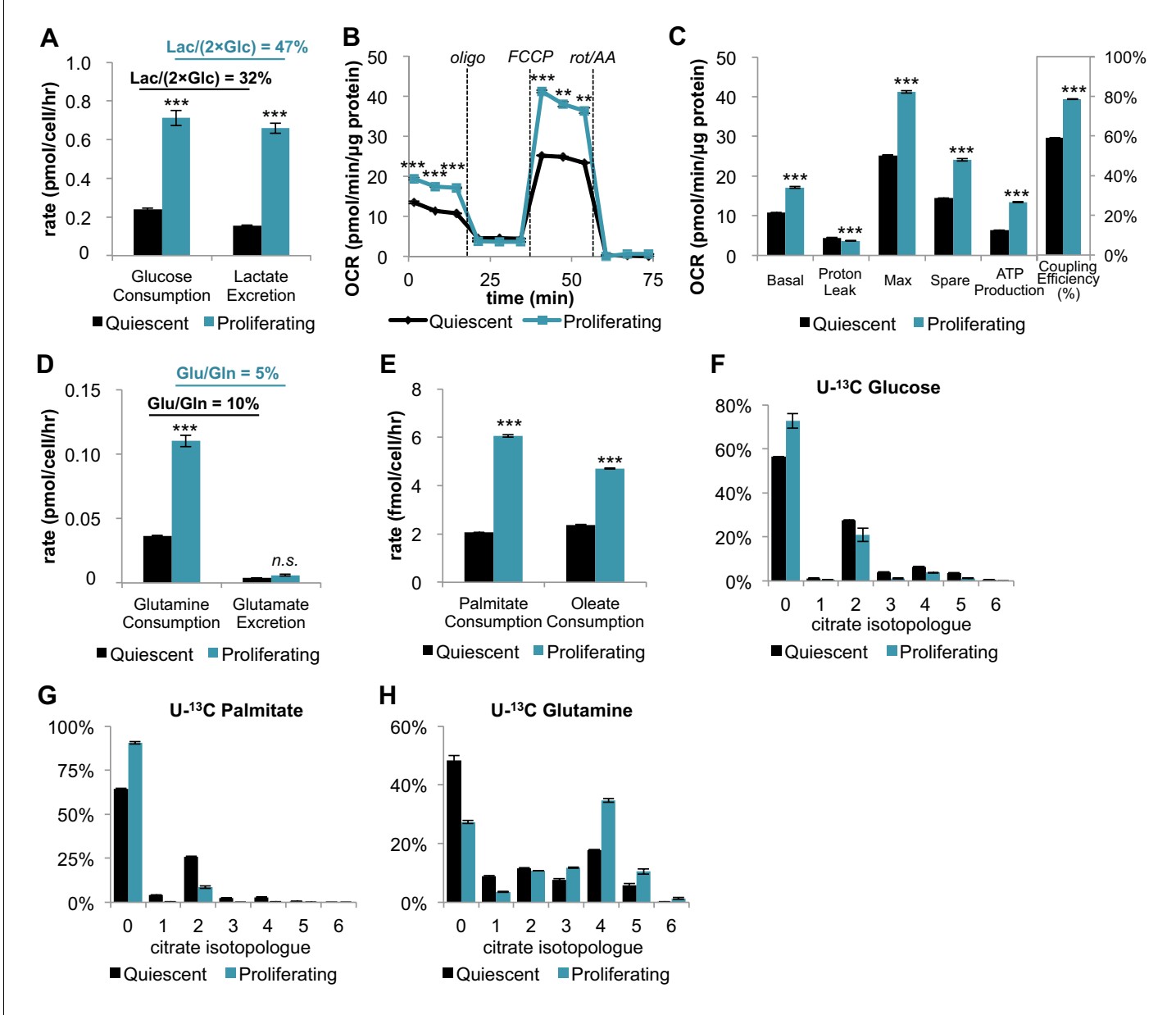

**Figure 1.** In addition to increasing glucose consumption and lactate excretion, proliferating fibroblasts also increase mitochondrial respiration and mitochondrial coupling efficiency. (**A**) Glucose consumption and lactate excretion rates for quiescent and proliferating fibroblasts (n = 4). As expected, proliferating cells exhibit an enhanced glycolytic phenotype that is consistent with the Warburg effect. (**B**) Mitochondrial stress test of quiescent and proliferating fibroblasts. OCR was normalized to protein amount to take into account differences in cell size. Displayed OCR values were corrected for non-mitochondrial respiration (n = 3). (**C**) Measured and calculated parameters of mitochondrial respiration (using results from *Figure 1B*). We note that the coupling efficiency is calculated as the ratio of the OCR required for ATP production relative to the basal OCR in the same sample and therefore is independent of the sample normalization method (n = 3). (**D**) Glutamine consumption and glutamate excretion rates for quiescent and proliferating fibroblasts (n = 4). (**E**) Palmitate and oleate consumption rates for quiescent and proliferating fibroblasts (n = 4). (**F–H**) Isotopologue distribution pattern of citrate after cells were labeled with U-[13]C glucose (**F**), U-[13]C palmitate (**G**), or U-[13]C glutamine (**H**) for 6 hr (n = 3). Data are presented as mean ±SEM. **p<0.01, ***p<0.001, *n.s.* not statistically significant. OCR, oxygen consumption rate; oligo, oligomycin; rot, rotenone; AA, Antimycin A.

DOI: https://doi.org/10.7554/eLife.41351.003

The following source data and figure supplements are available for figure 1:

**Source data 1.** Total accounting of glucose utilization in quiescent and proliferating cells.

DOI: https://doi.org/10.7554/eLife.41351.006

**Source data 2.** Labeling percentages of [13]C-enriched precursors for *Figure 1*.

DOI: https://doi.org/10.7554/eLife.41351.007

*Figure 1 continued on next page*

*Figure 1 continued*

**Source data 3.** Mass isotopologue distributions for all metabolites analyzed by LC-MS in *Figure 1F–H*.
DOI: https://doi.org/10.7554/eLife.41351.008
**Figure supplement 1.** Mitochondrial stress test of quiescent and proliferating fibroblasts normalized by cell number.
DOI: https://doi.org/10.7554/eLife.41351.004
**Figure supplement 2.** Proliferating fibroblasts increase their consumption rate of cystine by two-fold without altering the expression level of the cystine/glutamate antiporter SLC7A11.
DOI: https://doi.org/10.7554/eLife.41351.005

consistent with reports from other cells (*Fan et al., 2013*). Given that previous studies have shown that glutamine dependence is correlated with cystine uptake through the cystine/glutamate antiporter SLC7A11 (*Muir et al., 2017*), we examined whether these metabolite changes were enabled by transporter expression. Even though the level of SLC7A11 protein was unchanged, we observed more than a two-fold increase in cystine comsumption for proliferating fibroblasts compared to quiescent fibroblasts (*Figure 1—figure supplement 2*). Increased influx of cystine may drive the export of glutamate, thereby depleting the pool of intracellular glutamate/αKG and promoting glutamine anaplerosis (*Muir et al., 2017*).

## Increased oxidative phosphorylation during proliferation is supported by mitochondrial fusion

We next sought to understand how fibroblasts support increased OXPHOS during proliferation. We reasoned that one mechanism might be by increasing mitochondrial mass in the proliferative state. We used RT-PCR to determine the relative copy number of mitochondrial DNA (mtDNA) to genomic DNA (gDNA), from which we inferred mitochondrial mass. The mtDNA to gDNA ratio was unchanged between quiescent and proliferating cells (*Figure 2A*). Consistent with these data, we also observed no change in expression of respiratory enzymes, as determined by immunoblotting of electron transport chain (ETC) subunits (SDHA for complex II, cytochrome c for complex III, COX IV for complex IV, and ATP5A for complex V) (*Figure 2B*). Our results indicate that proliferating fibroblasts do not support elevated levels of OXPHOS by increasing mitochondrial mass or by increasing the expression of respiratory enzymes.

As an alternative, we then considered the possibility that fibroblasts regulate OXPHOS during proliferation by mitochondrial dynamics. Previous studies have shown that mitochondrial fusion is associated with an increased respiration rate in addition to an improved coupling efficiency (*Legros et al., 2002*; *Westermann, 2012*). To assess mitochondrial morphology, we applied electron microscopy (EM) imaging (*Figure 2C*) and fluorescence imaging (*Figure 2—figure supplement 1*). Both techniques showed that proliferating cells have elongated mitochondria, while mitochondria in quiescent cells were relatively short and fragmented. Quantitative analysis of 100 random mitochondria in each condition showed a statistically significant increase in the mitochondrial length of proliferating cells compared to quiescent cells (*Figure 2D*). In addition, we found that mitochondria in proliferating cells had a significantly higher level of mitochondrial fusion proteins (Mfn1, Mfn2, and OPA1) compared to mitochondria in quiescent cells (*Figure 2—figure supplement 2*). When we inhibited mitochondrial fusion by knocking down Mfn2, a protein required for the fusion of the outer mitochondrial membrane, the elongated mitochondrial phenotype in proliferating fibroblasts was suppressed (*Figure 2C–D*, *Figure 2—figure supplement 3*). To determine whether mitochondrial fusion is required for increased OXPHOS during proliferation, we compared the oxygen consumption rates of proliferating scrambled siRNA controls (SSC) to Mfn2 knockdowns (Mfn2$^{KD}$). We point out that these comparisons were conducted when cells were in the proliferating exponential growth phase. Notably, Mfn2 knockdowns had a statistically significant decrease in respiration rate, ATP production, and mitochondrial coupling efficiency (*Figure 2E–F*). Given that mitochondria in quiescent fibroblasts are already largely fragmented, as expected, the effect of Mfn2 knockdown on basal mitochondrial respiration was minimal in quiescent cells (*Figure 2—figure supplement 4*).

Independent of contact inhibition, cellular quiescence can also be achieved by serum starvation (*Yao, 2014*). By using serum starvation, we sought to extend our comparison of the proliferative and quiescent states to other cells. Consistent with our contact-inhibition results, we found that serum starved 3T3-L1 and HCT116 cells had fragmented mitochondria relative to the same cells in the non-

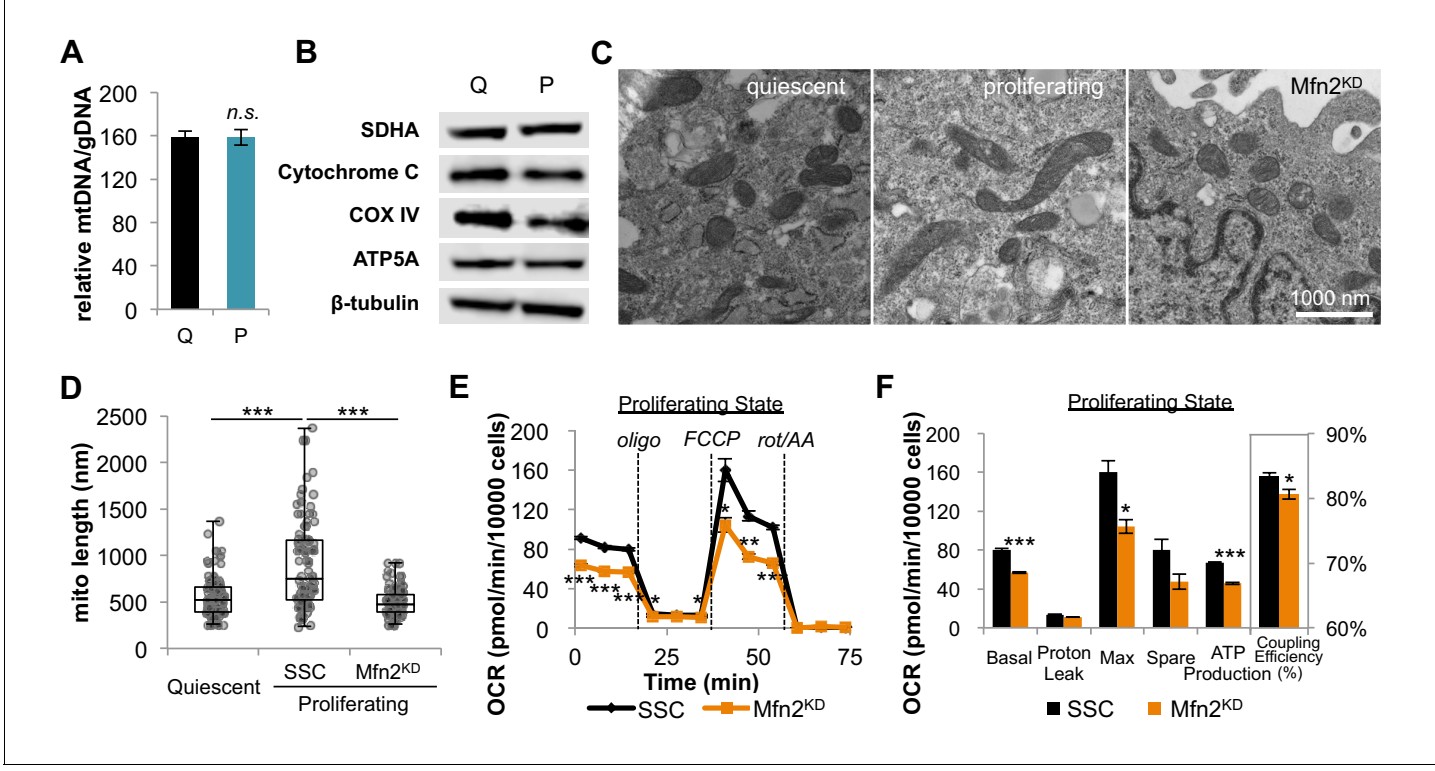

**Figure 2.** Proliferating fibroblasts regulate respiration by mitochondrial fusion. (A) Mitochondrial mass remains the same for quiescent (Q) and proliferating (P) fibroblasts as estimated by the ratio of mtDNA to gDNA (n = 3). (B) Proliferating fibroblasts have similar protein expression levels of ETC subunits as quiescent fibroblasts. (C) Representative EM images of mitochondria in quiescent fibroblasts, proliferating fibroblasts, and proliferating Mfn2 knockdowns show changes in mitochondrial elongation. (D) Statistical analysis of mitochondrial length in quiescent fibroblasts, proliferating fibroblasts, and proliferating Mfn2 knockdowns. For each condition, 100 random mitochondria were measured from EM images. Data are presented as mean ±SD. (E) Mitochondrial stress test of scrambled siRNA control (SSC) cells and Mfn2 knockdown cells (Mfn2$^{KD}$), both in the proliferating state (n = 3). Data are presented as mean ±SEM. (F) Measured and calculated parameters of mitochondrial respiration (using results from **Figure 2E**) (n = 3). Data are presented as mean ±SEM. *p<0.05, **p<0.01, ***p<0.001, *n.s.* not statistically significant. OCR, oxygen consumption rate; oligo, oligomycin; rot, rotenone; AA, Antimycin A.

DOI: https://doi.org/10.7554/eLife.41351.009

The following figure supplements are available for figure 2:

**Figure supplement 1.** Fluorescence imaging shows that mitochondria are elongated in proliferating fibroblasts but not in quiescent fibroblasts.
DOI: https://doi.org/10.7554/eLife.41351.010

**Figure supplement 2.** Mitochondria in proliferating fibroblasts have increased levels of mitochondrial fusion proteins compared to quiescent fibroblasts.
DOI: https://doi.org/10.7554/eLife.41351.011

**Figure supplement 3.** Immunoblot analysis shows that Mfn2 remained significantly knocked down during the course of the entire experiment.
DOI: https://doi.org/10.7554/eLife.41351.012

**Figure supplement 4.** Mfn2 knockdown has minimal effect on basal mitochondrial respiration in quiescent (Q) fibroblasts.
DOI: https://doi.org/10.7554/eLife.41351.013

**Figure supplement 5.** Serum starvation induces mitochondrial fragmentation in 3T3-L1 and HCT116 cells.
DOI: https://doi.org/10.7554/eLife.41351.014

starved state (**Figure 2—figure supplement 5**). Mitochondrial elongation occurred as soon as 3 hr after reintroducing serum and continued as cells exited the quiescent state (**Figure 2—figure supplement 5B**).

## Inhibiting mitochondrial fusion decreases proliferation by limiting aspartate synthesis

Having established that mitochondrial fusion increased in dividing cells, we wished to consider its effects on proliferation. Upon Mfn2 knockdown, we observed a ~ 30% decrease in proliferation rate

compared to scrambled siRNA controls (*Figure 3A*). Re-expressing siRNA-resistant Mfn2 (Mfn2^res) in Mfn2 knockdowns restored Mfn2 protein level, mitochondrial respiration, and cellular proliferation (*Figure 3—figure supplement 1*). When we overexpressed Mfn2 in wildtype 3T3-L1 fibroblasts, we observed a significant increase in both mitochondrial respiration and proliferation (*Figure 3—figure*

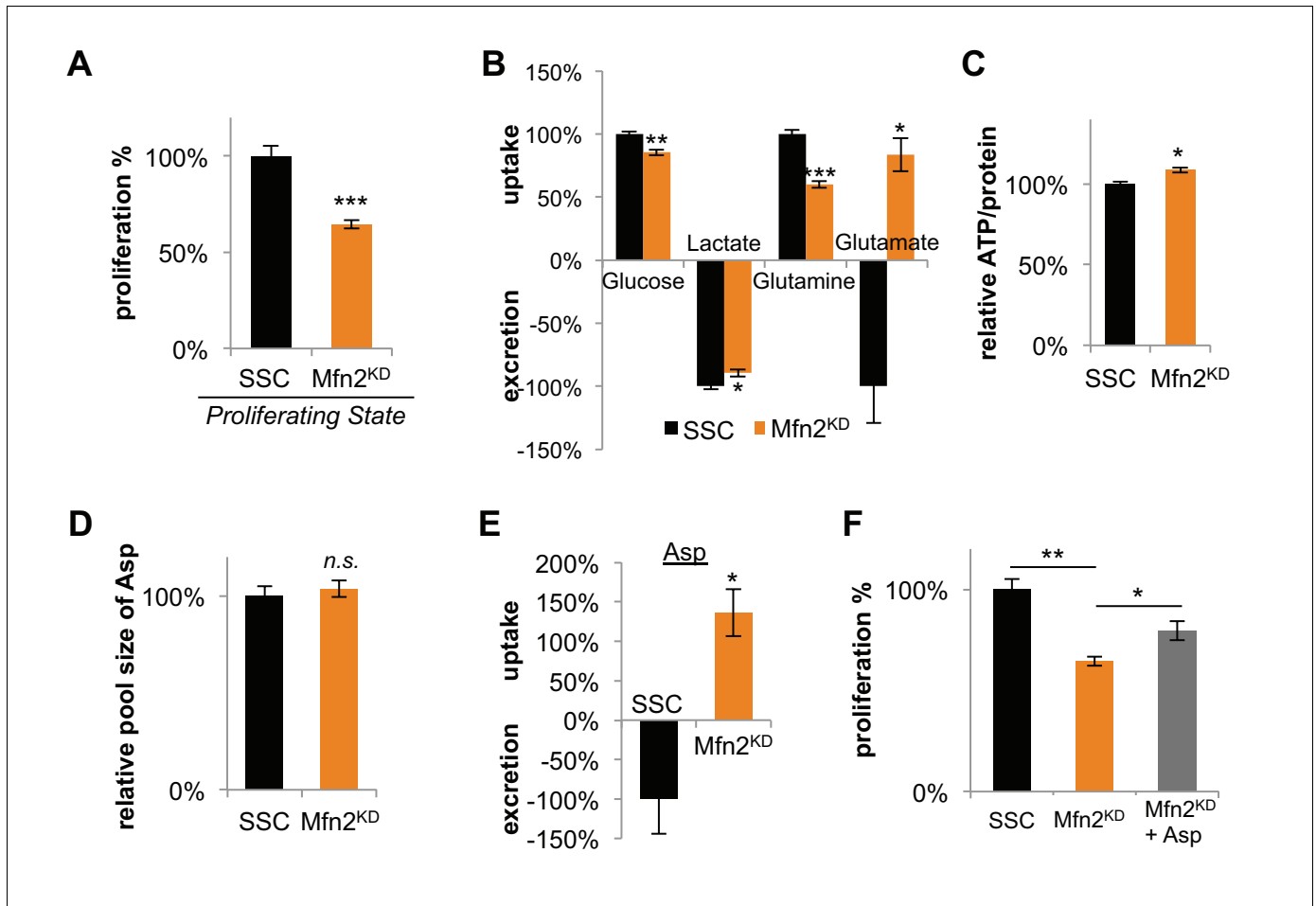

**Figure 3.** Inhibition of mitochondrial fusion by Mfn2 knockdown slows proliferation by limiting aspartate synthesis. (A) Proliferation was assessed by using a CyQUANT assay after cells were treated with scrambled siRNA control (SSC) or Mfn2 siRNA for 72 hr (n = 5). (B) Mfn2 knockdown alters nutrient utilization (n = 4). Glutamine consumption decreases in Mfn2 knockdown cells, which is consistent with a decreased demand for glutamine to fuel a reduced level of OXPHOS. (C) Intracellular ATP levels in scrambled siRNA controls (SSC) were lower relative to Mfn2 knockdowns. ATP luminescence signals were normalized to protein amount (n = 5). (D) The intracellular pool of aspartate remained unchanged upon Mfn2 knockdown (n = 3). Pool sizes were normalized by dry cell mass and internal standard. (E) Scrambled siRNA controls excrete aspartate into the media, while Mfn2 knockdowns uptake aspartate from the media (n = 4). (F) Supplementing the media with 1 mM aspartate partially rescued the proliferation of Mfn2 knockdowns (n = 6). Data are presented as mean ±SEM. *$p<0.05$, **$p<0.01$, ***$p<0.001$, *n.s.* not statistically significant.

DOI: https://doi.org/10.7554/eLife.41351.015

The following source data and figure supplements are available for figure 3:

**Source data 1.** Sequences for dicer-substrate short interfering RNA (DsiRNA) and siRNA-resistant Mfn2^res.
DOI: https://doi.org/10.7554/eLife.41351.019
**Figure supplement 1.** Expression of siRNA-resistant Mfn2 (Mfn2^res) in Mfn2 knockdowns restored Mfn2 protein level, mitochondrial respiration, and cellular proliferation.
DOI: https://doi.org/10.7554/eLife.41351.016
**Figure supplement 2.** Overexpression of Mfn2 in 3T3-L1 fibroblasts increases mitochondrial respiration and cellular proliferation.
DOI: https://doi.org/10.7554/eLife.41351.017
**Figure supplement 3.** Decreased labeling percentages of TCA cycle intermediates and decreased cystine consumption in Mfn2 knockdowns (Mfn2^KD) suggests a decrease in glutamine anaplerosis compared to scrambled siRNA controls (SSC).
DOI: https://doi.org/10.7554/eLife.41351.018

*supplement 2*). These data suggest that promoting mitochondrial fusion is sufficient to drive proliferation. We also observed considerable changes in nutrient utilization between Mfn2 knockdowns and scrambled siRNA controls that were consistent with decreased proliferation and reduced OXPHOS (*Figure 3B*). Mfn2 knockdowns decreased their glucose uptake by 15% and decreased their lactate excretion by 10%. More notably, knocking down Mfn2 caused cells to decrease glutamine consumption by 40%. Given the reduced rate of OXPHOS in Mfn2 knockdowns, these data are consistent with glutamine serving as a major carbon source for the TCA cycle. Tracing experiments confirmed a significant decrease in labeling of TCA cycle intermediates from U-$^{13}$C glutamine in Mfn2 knockdowns (*Figure 3—figure supplement 3A–C*). Consistent with decreased glutamine anaplerosis, Mfn2 knockdowns had a 40% decrease in cystine consumption (*Figure 3—figure supplement 3D*).

Our results show that mitochondrial fusion supports a high level of OXPHOS, which is needed to sustain rapid cellular proliferation. We speculated that the decrease in proliferation rate upon Mfn2 knockdown might be due to a shortage of energy from the decreased rate of OXPHOS. Surprisingly, however, we found that intracellular levels of ATP actually increased slightly in Mfn2 knockdowns relative to scrambled siRNA control cells (*Figure 3C*). This result suggested that cells could compensate for a reduced energy yield from OXPHOS, but that OXPHOS may serve another indispensable function in our knockdowns. Previous studies have shown that an essential role of OXPHOS in proliferating cells is to regenerate reducing equivalents in support of aspartate synthesis (*Birsoy et al., 2015*; *Sullivan et al., 2015*). Through reactions in the malate-aspartate shuttle, increased oxygen consumption may support a higher rate of aspartate generation. Although we did not observe a difference in the intracellular pool of aspartate between scrambled siRNA controls and Mfn2 knockdowns (*Figure 3D*), we did find a striking change in aspartate uptake. While the control cells excreted aspartate, the Mfn2 knockdown cells consumed aspartate from the media (*Figure 3E*). Moreover, the proliferation of Mfn2 knockdowns could be partially rescued by supplementing cell-culture media with aspartate (*Figure 3F*).

## H-*Ras* transformed fibroblasts exhibit higher mitochondrial respiration

Although 3T3-L1 fibroblasts are immortalized, unlike transformed cells, they remain sensitive to contact inhibition and retain the ability to differentiate. To evaluate whether transformed cells similarly rely on OXPHOS and mitochondrial fusion, we generated stable transfected 3T3-L1 cells expressing the oncogene H-*Ras* (G12V), a constitutively active mutant (*Figure 4A*). H-*Ras* transfected fibroblasts assumed a transformed morphology and their growth was no longer contact inhibited, with high-density cultures forming multiple layers of cells (*Figure 4B–C*). It was confirmed that the transformed cells did not undergo oncogene-induced senescence (*Figure 4—figure supplement 1*).

To study the effect of H-*Ras* transformation on mitochondrial metabolism, we compared the oxygen consumption rates of proliferating empty vector (EV) control cells in the exponential growth phase to H-*Ras* transformed (Ras) cells. Surprisingly, we found that Ras cells had a ~ 73% increase in basal respiration and a ~ 72% increase in ATP production compared to EV cells (*Figure 4D–E*). It is interesting to note that we found no difference in the mitochondrial coupling efficiencies between conditions (*Figure 4E*). This result is consistent with the observation that mitochondria are not further fused in Ras cells relative to proliferating EV controls (*Figure 4F*). Given that Ras cell mitochondria remain elongated to the same extent as EV controls, we next evaluated increased mitochondrial mass as a possible mechanism to support elevated levels of OXPHOS. For Ras cells, we observed a > 20-fold increase in the mRNA level of peroxisome proliferator activated receptor coactivator (*PGC1α*), a master regulator for mitochondrial biogenesis (*LeBleu et al., 2014*; *Scarpulla, 2011*) (*Figure 4G*). Ras cells had more mitochondrial mass and increased protein expression levels of ETC subunits (*Figure 4H–I*), suggesting that increased OXPHOS in Ras cells is driven by mitochondrial biogenesis. Activation of mitochondrial biogenesis upon *Ras* transformation did not change protein levels of components in the ERK/AMPK pathway or the KSR1 pathway, as has been previously suggested for other cells (*Figure 4—figure supplement 2*) (*Dard et al., 2018*; *Weinberg et al., 2010*). We wish to emphasize that even though Ras cell mitochondria are not further elongated, they remain fused to the same level as proliferating non-transformed fibroblasts. Therefore, when mitochondrial fusion was inhibited by knocking down Mfn2, the proliferation rates and OXPHOS of both EV cells and Ras cells were significantly attenuated (*Figure 4J*, *Figure 4—figure supplement 3*). In addition, the proliferation of various cancer cell lines also proved sensitive to Mfn2 knockdown

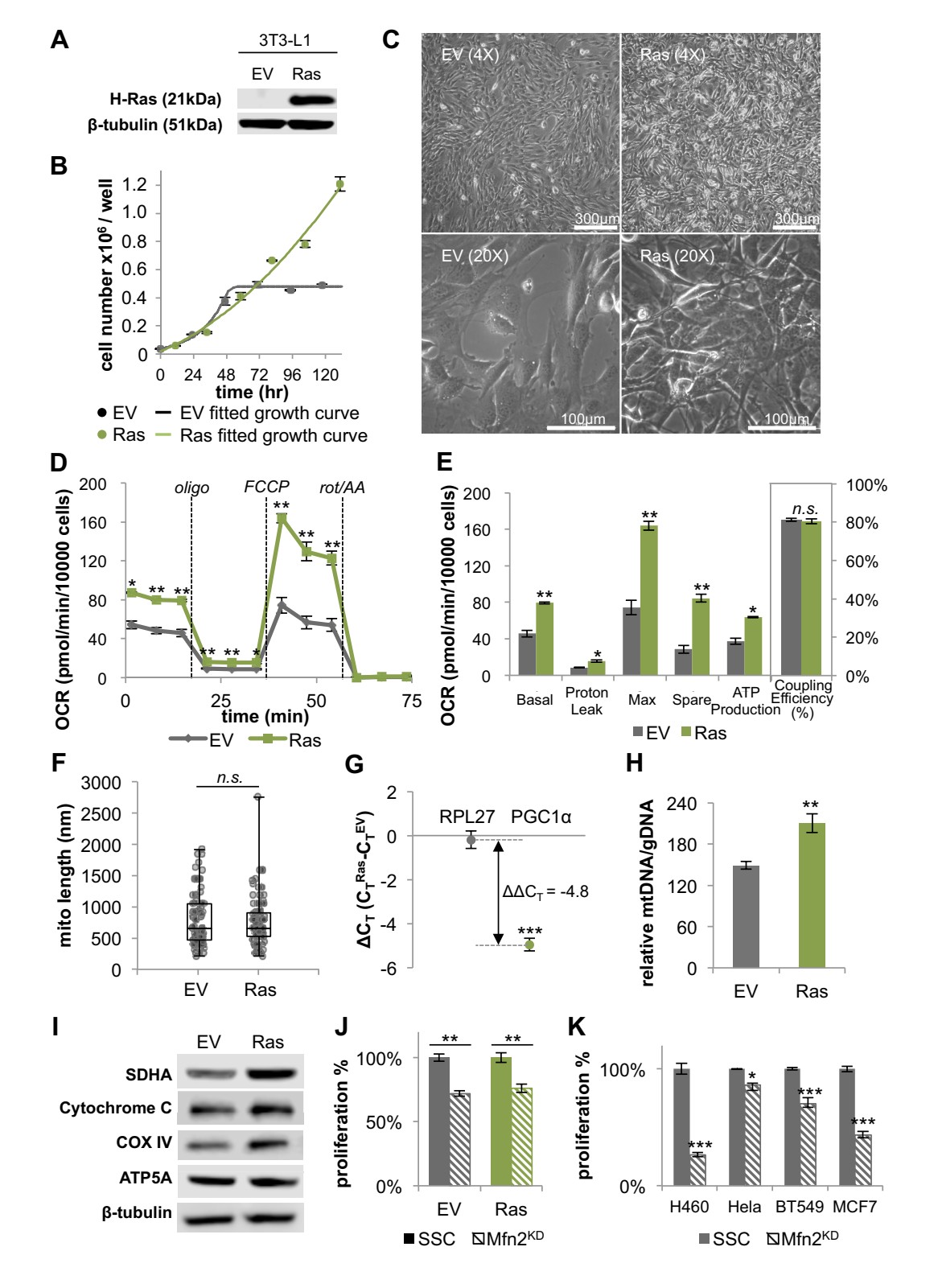

**Figure 4.** H-*Ras* transformed fibroblasts (Ras) have elongated mitochondria and increased OXPHOS that is supported by mitochondrial biogenesis. (**A**) Immunoblotting of whole-cell lysates shows H-*Ras* expression in transformed fibroblasts, but not in empty vector (EV) controls. (**B**) Growth curve shows the loss of contact inhibition in H-*Ras* transformed fibroblasts. The proliferation of EV controls remains cell-density dependent (n = 4). (**C**) Ras cells exhibit a morphological change and gain the ability to grow on top of each other to form multiple cell layers. (**D**) Mitochondrial stress test of EV

*Figure 4 continued on next page*

*Figure 4 continued*

controls and Ras cells (n = 3). The respiration of Ras cells is statistically increased compared to EV controls. (E) Measured and calculated parameters of mitochondrial respiration (using results from *Figure 4D*) (n = 3). (F) Statistical analysis of mitochondrial length in EV controls and Ras cells. In each condition, 100 random mitochondria were measured from EM images. Data are presented as mean ±SD. (G) RT-PCR shows that Ras cells have over a 20-fold increase in mRNA levels of *PGC1α* (n = 3). (H) Ras cells have increased mitochondrial mass, as indicated by an increased mtDNA/gDNA compared to EV controls (n = 3). (I) Immunoblotting of whole-cell lysates shows that Ras cells have higher expression levels of ETC subunits compared to EV controls. (J) Mfn2 knockdown decreases cellular proliferation in both EV controls and Ras cells. Proliferation was assessed by manual counting after cells were treated with scrambled siRNA control (SSC) or Mfn2 siRNA (Mfn2KD) for 72 hr. Relative proliferation was normalized to either an EV SSC or a RAS SSC for each condition (n = 4). (K) Mfn2 knockdown decreases cellular proliferation in H460 (lung cancer cells), HeLa (cervical cancer cells), BT549 (breast cancer cells), and MCF7 (breast cancer cells). Proliferation was assessed by using a CyQUANT proliferation assay after cells were treated with scrambled siRNA control (SSC) or Mfn2 siRNA (Mfn2KD) for 72 hr (n = 5). Unless specified, data are presented as mean ±SEM. *p<0.05, **p<0.01, ***p<0.001, *n.s.* not statistically significant. OCR, oxygen consumption rate; oligo, oligomycin; rot, rotenone; AA, Antimycin A.

DOI: https://doi.org/10.7554/eLife.41351.020

The following figure supplements are available for figure 4:

**Figure supplement 1.** β-Galactosidase staining shows that empty vector control (EV) and H-*Ras* transformed fibroblasts (Ras) do not undergo senescence.

DOI: https://doi.org/10.7554/eLife.41351.021

**Figure supplement 2.** Immunoblot analysis of components in the ERK/AMPK pathway and the KSR1 pathway shows no difference between empty vector controls (EV) and *Ras* transformed fibroblasts (Ras).

DOI: https://doi.org/10.7554/eLife.41351.022

**Figure supplement 3.** Inhibiting mitochondrial fusion by Mfn2 knockdown decreases mitochondrial respiration rates in EV controls and Ras cells.

DOI: https://doi.org/10.7554/eLife.41351.023

**Figure supplement 4.** H-*Ras* transformed fibroblasts have increased metabolic activity, elevated levels of ROS, and DNA damage.

DOI: https://doi.org/10.7554/eLife.41351.024

(*Figure 4K*). We conclude that mitochondrial fusion is important to sustain cellular proliferation, independent of oncogenic transformation.

## Transformation with H-*Ras* increases oxidative stress and DNA damage

We speculated that constitute activation of Ras might enhance metabolic phenotypes we associated with proliferation in non-transformed cells. Indeed, in addition to elevated OXPHOS, Ras cells consumed more glucose, glutamine, and fatty acids relative to EV controls (*Figure 4—figure supplement 4A*). Given that Ras cells do not proliferate faster than EV cells (*Figure 4B*), the increase in metabolic activity is unlikely due to proliferative demand but rather associated with Ras signaling activation. Consistent with the notion that energy is not limiting during proliferation (*Locasale and Cantley, 2011*), we found that Ras cells had a higher intracellular level of ATP compared to EV controls (*Figure 4—figure supplement 4B*). In addition to increased ATP production, elevated OXPHOS activity in Ras cells also contributed to higher levels of reactive oxygen species (ROS) (*Figure 4—figure supplement 4C*). The associated oxidative stress could be buffered by treating Ras cells with the antioxidant N-acetyl cysteine (NAC). Alternatively, oxidative stress could be further induced by treating Ras cells with bromodeoxyuridine (BrdU), a thymidine analog (*Figure 4—figure supplement 4C*). We suspected that the elevated levels of ROS in Ras cells may lead to increased DNA damage. We verified this to be the case by showing that Ras cells had increased phosphorylation on histone H2A.X (Ser139), which has been suggested as a sensitive marker for DNA damage (*Sharma et al., 2012*) (*Figure 4—figure supplement 4D*). These findings suggest that the increase in OXPHOS upon constitutive Ras activation leads to elevated oxidative stress and DNA damage, while not directly contributing to the anabolic demands of proliferation. Given the pleiotropic effects of *Ras* and other oncogenes, the generality of these findings to additional cell types will require further investigation.

## Discussion

Most proliferating cells assume a metabolic phenotype known as the Warburg effect (*Lunt and Vander Heiden, 2011*). Although the enhanced glycolytic characteristics of the Warburg effect are generally well established, metabolic changes associated with mitochondria have proven more challenging to interrogate. In part, this is because proliferation has largely been studied in the

context of cancer where some experimental factors are complicated to control (e.g. tumor microenvironment, oxygen availability, genetics, proliferation, etc.). Here, we applied a well-controlled fibroblast model to quantify changes in mitochondrial respiration that occur in quiescent cells, non-transformed proliferating cells, and transformed proliferating cells.

Strikingly, despite the frequent assumption that increased glycolysis in cells exhibiting the Warburg effect is associated with decreased OXPHOS, we found that mitochondrial respiration increased by nearly two-fold in non-transformed proliferating cells relative to quiescent cells. Moreover, mitochondrial respiration increased by nearly another factor of two when the cells were transformed with H-*Ras*. We wish to point out that the regulatory mechanism for increasing respiration between quiescent and non-transformed proliferating cells was different from that between non-transformed and transformed cells. The quiescent to proliferating transition was supported by mitochondrial fusion without any increase in mitochondrial mass, whereas the non-transformed to transformed transition was supported by mitochondrial biogenesis without any further increase in mitochondrial elongation. Similar increases in mitochondrial biogenesis and OXPHOS upon Ras activation have been reported in other cell lines (*Funes et al., 2007*; *Moiseeva et al., 2009*; *Telang et al., 2007*). In addition to Ras, various other signaling pathways that regulate cellular proliferation (e.g. c-Myc and mTOR) have also been found to activate mitochondrial biogenesis (*Vyas et al., 2016*). Notably, however, our data suggest that the proliferation rate of both non-transformed and transformed fibroblasts is dependent on mitochondrial fusion.

It is interesting to consider why OXPHOS is increased by mitochondrial fusion in proliferating cells. Since ATP levels actually increased when OXPHOS was impaired by Mfn2 knockdown,

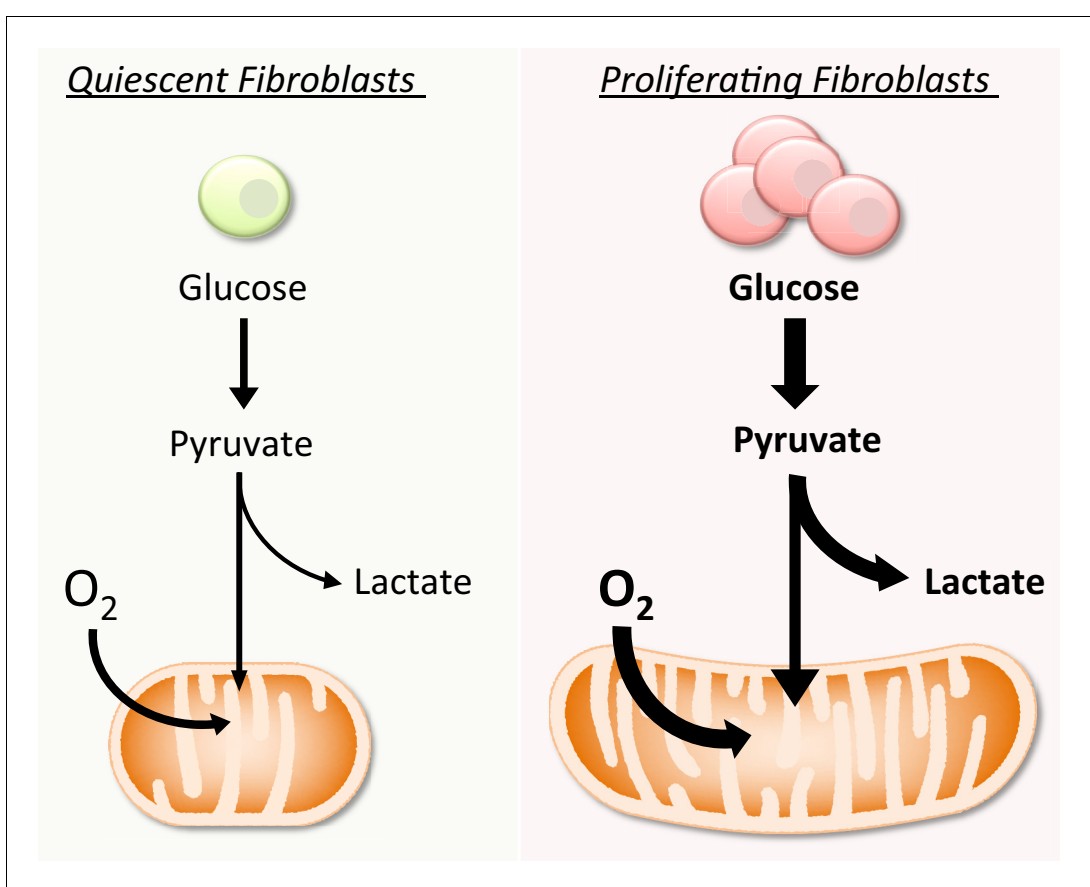

**Figure 5.** Schematic representation of the metabolic differences between quiescent and proliferating fibroblasts. Compared to quiescent fibroblasts, proliferating cells increase both glycolysis and OXPHOS. The increase in OXPHOS is supported by mitochondrial fusion.
DOI: https://doi.org/10.7554/eLife.41351.025

mitochondrial fusion does not seem to be required to support energetic demands. Instead, increased respiration during proliferation seems to be needed to recycle reducing equivalents in support of aspartate synthesis. Only after mitochondrial fusion was inhibited did cells start consuming aspartate from the media. Moreover, proliferation could be partially restored in Mfn2 knockdowns by supplementing cell media with aspartate. Building on previous studies (*Birsoy et al., 2015*; *Sullivan et al., 2015*), these data suggest not only that respiration is required to meet aspartate demands, but that a high level of OXPHOS may be necessary to fulfill this role. On the other hand, too much OXPHOS may be detrimental to cells. Increasing OXPHOS beyond the level observed in non-transformed proliferating fibroblasts with the H-*Ras* oncogene did not increase the rate of proliferation. Instead, the considerably higher levels of OXPHOS induced by mitochondrial biogenesis resulted in oxidative stress and DNA damage. These results suggest that, unlike the mechanisms that increase mitochondrial respiration in normal proliferating cells, Ras activation may promote additional malignant transformations by creating genomic instability (*Tubbs and Nussenzweig, 2017*).

Despite the association between the Warburg effect and rapid proliferation, a rationalization for the preference of glycolysis over OXPHOS has proven elusive (*Liberti and Locasale, 2016*). A challenge has been explaining how the switch to a metabolic program that is less energetically efficient supports the synthetic burden of cell replication. Transformation of glucose to lactate yields only two moles of ATP per mole of glucose, whereas complete oxidation of glucose by the TCA cycle yields ~ 32 moles of ATP per mole of glucose. Hypotheses have emerged that proliferating cells sacrifice ATP yield from glucose for other advantages such as a high rate of ATP production, decreased volume of enzymatic machinery, or increased concentrations of macromolecular precursors (*Lunt and Vander Heiden, 2011*; *Slavov et al., 2014*; *Vazquez et al., 2010*). Yet, to date, the benefits of using glycolysis over OXPHOS for proliferation have remained controversial. In this study, we provide evidence that the Warburg effect does not necessitate decreased OXPHOS. Rather, in the cells we examined here, glycolysis and OXPHOS are both elevated by significant levels during proliferation (*Figure 5*). Thus, the need to rationalize a preference for glycolysis over OXPHOS during proliferation may be unnecessary.

# Materials and methods

## Key resources table

| Reagent type (species) or resource | Designation | Source or reference | Identifiers | Additional information |
|---|---|---|---|---|
| Cell line (*M. musculus*) | 3T3-L1 | American Type Culture Collection | RRID:CVCL_0123 | |
| Recombinant DNA reagent | Mfn2$^{res}$ | this study | | codon-optimized cDNA of Mfn2 in pcDNA3.1(+) vector (siRNA-resistant) |
| Recombinant DNA reagent | GFP | Genscript | | pcDNA3.1_N-eGFP |
| Recombinant DNA reagent | pCMV-VSV-G | Washington University | | Addgene plasmid # 8454 |
| Recombinant DNA reagent | pCMVDR8.2 | Washington University | | Addgene plasmid # 12263 |
| Recombinant DNA reagent | pLVX-HRasV12-hygromycin | Washington University | | |
| Antibody | Rabbit anti-Mfn2, monoclonal | Cell Signaling | Cat.#: 9482, RRID:AB_2716838 | WB (1:1000) |
| Antibody | Rabbit anti-OPA1, monoclonal | Cell Signaling | Cat.#: 80471 | WB (1:1000) |
| Antibody | Rabbit anti-COXIV, monoclonal | Cell Signaling | Cat.#: 4850, RRID:AB_2085424 | WB (1:1000) |

*Continued on next page*

*Continued*

| Reagent type (species) or resource | Designation | Source or reference | Identifiers | Additional information |
|---|---|---|---|---|
| Antibody | Rabbit anti-Ras (G12V), monoclonal | Cell Signaling | Cat.#: 14412, RRID:AB_2714031 | WB (1:1000) |
| Antibody | Rabbit anti-phospho-H2A.X, monoclonal | Cell Signaling | Cat.#: 9718, RRID:AB_2118009 | WB (1:1000) |
| Antibody | Rabbit anti-AMPKα, monoclonal | Cell Signaling | Cat.#: 5831, RRID:AB_10622186 | WB (1:1000) |
| Antibody | Rabbit anti-KSR1,, polyclonal | Cell Signaling | Cat.#: 4640 | WB (1:1000) |
| Antibody | Rabbit anti-p44/42 MAPK (Erk1/2), monoclonal | Cell Signaling | Cat.#: 4695 | WB (1:1000) |
| Antibody | Rabbit anti-Phospho-p44/42 MAPK (Erk1/2), monoclonal | Cell Signaling | Cat.#: 4370 | WB (1:1000) |
| Antibody | Rabbit anti-EAAT1, monoclonal | Cell Signaling | Cat.#: 5684 | WB (1:1000) |
| Antibody | Rabbit anti-EAAT2, polyclonal | Cell Signaling | Cat.#: 3838 | WB (1:1000) |
| Antibody | Rabbit anti-EAAT3, monoclonal | Cell Signaling | Cat.#: 14501 | WB (1:1000) |
| Antibody | Rabbit anti-β-tubulin (HRP conjugated), monoclonal | Cell Signaling | Cat.#: 5346 | WB (1:1000) |
| Antibody | Mouse anti-SDHA, monoclonal | Santa Cruz Biotechnology | Cat.#: sc-166909, RRID:AB_10611174 | WB (1:500) |
| Antibody | Mouse anti-Cytochrome c, monoclonal | Santa Cruz Biotechnology | Cat.#: sc-13156, RRID:AB_627381 | WB (1:1000) |
| Antibody | Mouse anti-ATP5A, monoclonal | Santa Cruz Biotechnology | Cat.#: sc-136178 | WB (1:500) |
| Antibody | Mouse anti-Mfn1, monoclonal | Invitrogen | Cat.#: MA5-24789, RRID:AB_2717262 | WB (1:1000) |
| Antibody | Rabbit anti-SLC7A11, polyclonal | Invitrogen | Cat.#: PA1-16893, RRID:AB_2286208 | WB (1:1000) |
| Antibody | Mouse anti-PDH, monoclonal | Invitrogen | Cat.#: 459400, RRID:AB_2532238 | WB (1:1000) |
| Antibody | Goat anti-Rabbit | LiCor | Cat.#: 926–80011, RRID:AB_2721264 | WB (1:5000) |
| Antibody | Goat anti-Mouse | LiCor | Cat.#: 926–80010 | WB (1:5000) |
| Sequence-based reagent | Mfn2 siRNA | Intergrated DNA Technologies | | TriFECTa DsiRNA Kit |
| Commercial assay or kit | CyQUANT proliferation assay | Thermo Fisher | Cat.#: C7026 | |
| Commercial assay or kit | BCA protein assay | Thermo Fisher | Cat.#: 23225 | |
| Commercial assay or kit | NovaQUANT mouse mitochondrial to nuclear DNA ratio kit | Millipore Sigma | Cat.#: 72621 | |
| Commercial assay or kit | ATP luminescence detection assay kit | Cayman Chemical | Cat.#: 700410 | |

*Continued on next page*

*Continued*

| Reagent type (species) or resource | Designation | Source or reference | Identifiers | Additional information |
|---|---|---|---|---|
| Commercial assay or kit | DCFDA assay | Cayman Chemical | Cat.#: 601520 | |

## Cell culture, growth curve, and proliferation assays

3T3-L1 cells were obtained from ATCC. H460, Hela, BT549, and MCF7 cells were obtained from Washington University. All cells were found to be negative for mycoplasma contamination. All cells were cultured in high-glucose DMEM (Life Technologies) containing 10% fetal bovine serum (FBS) (Life Technologies) and 1% penicillin/streptomycin (Life Technologies) at 37°C with 5% $CO_2$. To establish a growth curve, cells were collected every 12–24 hr and counted in trypan blue with an automated cell counter (Nexcelom). For assessing proliferation, cells were grown under various experimental conditions for 48–72 hr, and proliferation was determined by manual cell counting or by using a CyQUANT proliferation assay (Thermo) according to the manufacturer's instructions. For serum starvation, cells were cultured in DMEM (without FBS) for 48 hr. Proliferation was induced by transferring cells to media containing serum (20% FBS).

## Oxygen consumption assays

The oxygen consumption rate (OCR) of whole cells was determined by using the Seahorse XFp Extracellular Flux Analyzer (Seahorse Bioscience). Cells were trypsinized and plated on a miniplate 24 hr prior to the Seahorse assay. For Mfn2 knockdowns, cells were treated with scrambled siRNA as a control or Mfn2 siRNA for 48 hr prior to seeding. The assay medium consisted of 25 mM glucose, 4 mM glutamine, 50 µM palmitate-BSA, and 50 µM oleate-BSA in Seahorse base medium. The OCR was monitored upon serial injections of oligomycin (oligo, 2 µM), FCCP (1 µM), and a rotenone/antimycin A mixture (rot/AA, 1 µM). A concentration of 1 µM FCCP was determined to be optimal in separate experiments. OCR was normalized to the final cell number or total protein amount as determined by manual cell counting or by using a BCA assay, respectively. Data presented have been corrected for non-mitochondrial respiration.

## Palmitate, glucose, and glutamine labeling experiments and pool-size measurements

3T3-L1 fibroblasts were plated at ~20% confluence or 100% confluence to establish the proliferating or quiescent condition, respectively. EV controls and H-*Ras* transformed fibroblasts were plated at ~40% confluence. Then, 24 hr later, the medium was changed to medium in which natural-abundance glucose was replaced with U-$^{13}$C glucose or natural-abundance glutamine was replaced with U-$^{13}$C glutamine. For palmitate labeling, after 24 hr the medium was changed to medium containing 100 µM U-$^{13}$C palmitate-BSA and 100 µM natural abundance oleate-BSA. After labeling for 6 hr, cells were harvested, extracted, and analyzed as previously described (*Yao et al., 2018*). The polar portion of the extract was separated by using a Luna aminopropyl column (Phenomenex) coupled to an Agilent 1260 capillary HPLC system. The Luna column was used with the following buffers and linear gradient: A = 95% water, 5% acetonitrile (ACN), 10 mM ammonium hydroxide, 10 mM ammonium acetate; B = 95% ACN, 5% water. Mass spectrometry detection was carried out on an Agilent 6540 or 6545 Q-TOF coupled with an ESI source operated in negative mode. The identity of each metabolite was confirmed by comparing retention times and MS/MS data with standard compounds. The isotopologue distribution pattern was calculated by normalizing the sum of all isotopologues to 100%. The labeling percentages of tracers at the end of the experiments are presented in *Figure 1— source data 2*. Data shown have been corrected for natural abundance and isotope impurity (see *Figure 1—source data 3* for raw data). Pool sizes were calculated as the sum of all isotopologues and normalized to dry cell mass (measured by using an analytical balance) as well as a D8-phenylalanine internal standard.

## Nutrient-uptake analysis

After incubating cells in fresh media for 24 hr, the spent media were collected and analyzed. Known concentrations of isotope-labeled internal standards (glucose, lactate, glutamine, glutamate, palmitate, and aspartic acid; Cambridge Isotopes) were spiked into media samples before extraction. Extraction was performed with glass as previously reported (Yao et al., 2016b). Samples were measured by LC/MS analysis, with the method described above. The absolute concentration of each compound was determined by calculating the ratio between the fully unlabeled peak from samples and the fully labeled peak from standards. The consumption rates ($x$) were normalized by cell growth over the experimental time period by using the following equation where $N_0$ represents the starting cell number, $t$ represents incubation time, $DT$ represents doubling time, and $Y$ represents nutrient utilization.

$$Y = \int_0^t x \cdot N_0 \cdot 2^{t/DT} \cdot dt$$

## Real-time PCR analysis of mtDNA/gDNA and PGC1α−1 expression

DNA was extracted by using QuickExtract DNA extraction solution (Epicentre) according to the manufacturer's instructions. The ratio of mitochondrial DNA (mtDNA) to genomic DNA (gDNA) was measured by using a NovaQUANT mouse mitochondrial to nuclear DNA ratio kit (Millipore) with RT-PCR (Applied Biosystems). We applied the following expressions: $\Delta C_T = C_T^{Mitochondrial}\ C_T^{Nucleic}$ and mtDNA/gDNA = $2^{-\Delta CT}$. For measuring PGC1α−1 expression levels, RNA was extracted using Trizol (Invitrogen). cDNA was synthesized using Super-Script III First-Strand Synthesis SuperMix (Invitrogen). Amplifications were run with RT-PCR by using premade primers (IDT). The results were normalized to a housekeeping gene, RPL27. The following expressions were applied: $\Delta\Delta C_T = \Delta C_T^{PGC1\alpha-1} - \Delta C_T^{RPL27}$ and fold change = $2^{-\Delta\Delta CT}$.

## Knockdown and overexpression of Mfn2

Mfn2 silencing was achieved by using a validated pool of siRNA duplexes directed against mouse Mfn2 (TriFECTa Kit, IDT) and Lipofectamine RNAiMAX Transfection Reagents (Invitrogen) according to the manufacturer's instructions (see *Figure 3—source data 1* for the dicer-substrate short interfering RNA, DsiRNA, sequence). Cells given scrambled siRNA were used as a negative control. To rescue siRNA knockdowns, a siRNA-resistant cDNA that expresses wildtype Mfn2 was cloned in the pcDNA3.1+vector (GenScript) under a constitutive CMV promoter. The codon was optimized to be resistant to the siRNA added (see *Figure 3—source data 1* for cDNA sequence). The control vector was pcDNA3.1+N eGFP (GenScript), which expresses GFP instead of Mfn2. For rescue experiments, cells were first knocked down with siRNA for 12 hr and then transfected with plasmids using Lipofectamine 3000 (Invitrogen) for 36–48 hr. For overexpression, wildtype cells were transfected with plasmids for 36–48 hr.

## Immunoblot analysis

Cells or isolated mitochondria were lysed with RIPA buffer (Thermo Fisher Scientific) in the presence of a protease inhibitor and a phosphatase inhibitor cocktail (Thermo Fisher Scientific). Lysates were separated by SDS–PAGE under reducing conditions, transferred to a nitrocellulose membrane, and analyzed by immunoblotting. For primary and secondary antibodies, please refer to the Key Resources Table. β-tubulin was used as a loading control. Signal was detected with a WesternSure premium chemiluminescent substrate and the C-Digit Blot Scanner (LI-COR) according to the manufacturer's instructions.

## Mitochondrial length measurements with transmission electron microscopy

Samples were fixed in 2% paraformaldehyde/2.5% glutaraldehyde (Polysciences) in 100 mM sodium cacodylate buffer, pH 7.2 for 1 hr at room temperature. Samples were washed in sodium cacodylate buffer and postfixed in 1% osmium tetroxide (Polysciences) for 1 hr. Next, samples were rinsed in dH$_2$O prior to en bloc staining with 1% aqueous uranyl acetate (Ted Pella) for 1 hr. Following several rinses in dH$_2$O, samples were dehydrated in a graded series of ethanol and embedded in Eponate 12 resin (Ted Pella). Sections of 95 nm were cut with a Leica Ultracut UCT ultramicrotome (Leica

Microsystems), stained with uranyl acetate and lead citrate, and viewed on a JEOL 1200 EX transmission electron microscope (JEOL USA) equipped with an AMT eight megapixel digital camera and AMT Image Capture Engine V602 software (Advanced Microscopy Techniques). The length of 100 random mitochondria for each condition were measured and plotted.

### Confocal fluorescence microscopy

After removing the media, cells were incubated with 100 nM MitoTracker Red CMXRos (Thermo Fisher Scientific) dissolved in complete medium at 37°C for 30 min. Nuclei were stained with Hoechst 33342 (Thermo Fisher Scientific). Cells were imaged alive using a Zeiss LSM 880 confocal microscope equipped with Airyscan. Images were acquired with a Zeiss 20x, 40x, 63x/1.4 NA objective by using the ZEN Black acquisition software. Samples were excited with 405 and 543 nm laser lines. Images were processed and prepared with the ZEN Black software.

### Lentivirus production and *Ras* transformation

To generate lentivirus carrying oncogenic *Ras* (HRAS$^{V12}$), HEK293T cells were co-transfected with pCMV-VSV-G (a gift from Bob Weinberg; Addgene plasmid # 8454), pCMVΔR8.2 (a gift from Didier Trono; Addgene plasmid # 12263), and pLVX-HRas$^{V12}$-hygromycin (a gift from David Piwnica-Worms) constructs with Lipofectamine 2000 reagent (Invitrogen). Cell media were replaced with fresh growth media 24 hr after transfection. Supernatants with lentivirus were collected after a 24-hr incubation period. Collected lentivirus (5 mL) was used to infect $10^6$ 3T3-L1 fibroblast cells in the presence of 10 µg/ml protamine sulfate overnight for ~16 hr. Selection of HRAS$^{V12}$-expressing 3T3-L1 cells was achieved by 100 µg/ml hygromycin. *Ras* expression was verified by immunoblotting. Senescence was tested by using the β-galactosidase staining kit (Cell Signaling) according to the manufacturer's instructions.

### Intracellular ATP measurements

Intracellular ATP was measured by using an ATP luminescence detection assay kit (Cayman) according to the manufacturer's instructions. The signal was normalized by protein amount as determined by using a BCA assay (Pierce).

### Intracellular ROS measurements

Cells were treated with 5 mM N-acetyl cysteine (NAC) or 5-bromo-2'-deoxyuridine (BrdU) for 48 hr. Intracellular reactive oxygen species (ROS) were measured by using a DCFDA assay (Cayman) according to the manufacturer's instructions. The signal was normalized by protein amount as determined by using a BCA assay (Pierce).

## Acknowledgements

We thank Wandy Beatty and the Molecular Microbiology Core at Washington University for their support in imaging cells and mitochondria. We thank Gao-Yuan Liu for his discussion in experimental design. We thank Xiangfeng Niu for preparing the mitochondrial isolation solution.

## Additional information

#### Competing interests

Gary J Patti: GJP is a scientific advisory board member for Cambridge Isotope Laboratories and a recipient of the Agilent Early Career Professor Award. The other authors declare that no competing interests exist.

#### Funding

| Funder | Grant reference number | Author |
|---|---|---|
| National Institutes of Health | R35ES028365 | Gary J Patti |
| National Institutes of Health | R24OD024624 | Gary J Patti |

| National Institutes of Health | U01CA235482 | Gary J Patti |
| Pew Charitable Trusts | | Gary J Patti |
| Edward Mallinckrodt, Jr. Foundation | | Gary J Patti |

The funders had no role in study design, data collection and interpretation, or the decision to submit the work for publication.

## Author contributions
Cong-Hui Yao, Conceptualization, Data curation, Formal analysis, Investigation, Methodology, Writing—original draft, Project administration, Writing—review and editing; Rencheng Wang, Conceptualization, Methodology, Writing—review and editing; Yahui Wang, Project administration, Writing—review and editing; Che-Pei Kung, Resources, Methodology, Project administration, Writing—review and editing; Jason D Weber, Resources, Supervision, Writing—review and editing; Gary J Patti, Conceptualization, Supervision, Funding acquisition, Investigation, Methodology, Writing—original draft, Writing—review and editing

## Author ORCIDs
Cong-Hui Yao (iD) http://orcid.org/0000-0003-3922-1874
Che-Pei Kung (iD) http://orcid.org/0000-0002-1150-4998
Gary J Patti (iD) https://orcid.org/0000-0002-3748-6193

## Decision letter and Author response
Decision letter https://doi.org/10.7554/eLife.41351.031
Author response https://doi.org/10.7554/eLife.41351.032

# Additional files

## Supplementary files
• Transparent reporting form
DOI: https://doi.org/10.7554/eLife.41351.026

## Data availability
All data generated or analysed during this study are included in the manuscript and supporting files. Source data files have been provided for Figure 1F-H and sequences of DsiRNA as well as siRNA resistant Mfn2.

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
