## [Decision Letter]

Thank you for submitting your article "Mitochondrial Fusion Supports Increased Oxidative Phosphorylation during Cell Proliferation" for consideration by *eLife*. Your article has been reviewed by three peer reviewers, including Ralph DeBerardinis as the Reviewing Editor and Reviewer #1, and the evaluation has been overseen by a Reviewing Editor and Andrea Musacchio as the Senior Editor.

The reviewers have discussed the reviews with one another and the Reviewing Editor has drafted this decision to help you prepare a revised submission.

Summary:

This paper compares mitochondrial function and dynamics between proliferating and non-proliferating fibroblasts. The authors find that proliferation activates glycolysis and respiration concurrently. The gain in respiration is associated with enhanced glutamine-dependent labeling of TCA cycle intermediates and enhanced mitochondrial fusion and is reversed by silencing the fusion factor Mfn2. Blocking Mfn2 also reduces cell proliferation, and a compound reported to induce mitochondrial fusion modestly increases respiration. Expressing an oncogenic allele of HRAS induces both cell proliferation and respiration, with the latter effect being related to an increase in mitochondrial mass rather than a specific effect of fusion. The authors conclude that enhanced mitochondrial function, via mitochondrial fusion, supports fibroblast proliferation. Although some of these points have been made elsewhere in the literature, overall the paper provides a clear and convincing view of the metabolic transitions that accompany and enable mammalian cell proliferation, at least in this system.

Essential revisions:

1) Perhaps the most interesting question is how proliferating cells induce mitochondrial fusion. Can the authors provide an explanation for this switch, or at least better characterize how quickly this switch occurs when cells begin to proliferate?

2) One could argue that Mfn2 loss decreases OXPHOS overall rather than specifically preventing OXPHOS gain during cell proliferation. Can the authors show whether Mfn2 loss also decreases OXPHOS in quiescent cells?

3) Because the main readout of Mfn2 silencing is reduced proliferation (i.e. loss of fitness), the key experiments need to be controlled for off-target toxicity by re-expressing an siRNA-resistant cDNA of Mfn2.

4) The M1 experiment is important because it would indicate that mitochondrial fusion is sufficient to drive proliferation. But more data are needed. Does this dose enhance fusion in these cells? Does it cause metabolic changes consistent with enhanced fusion? Is the effect on cell proliferation dose-dependent?

5) Any insights into how HRAS induces PGC1α would improve the paper. The authors should also cite Weinberg et al. (2010) which first reported the role of oncogenic HRAS and KRAS on ROS and respiration.

6) Are some of these metabolic changes enabled by changes in membrane transporter expression? Assessing the levels of SLC7A11 (cystine-driven glutamine anaplerosis) and SLC1A2/3 (glutamate/aspartate uptake) in the quiescent and proliferating states with and without Mfn2 expression might help explain some of these changes.

---

## [Author Response]

Essential revisions:1) Perhaps the most interesting question is how proliferating cells induce mitochondrial fusion. Can the authors provide an explanation for this switch, or at least better characterize how quickly this switch occurs when cells begin to proliferate?

We have performed experiments to better understand how proliferating cells support mitochondrial fusion and to understand how quickly the switch occurs as cells begin to proliferate. Our results are consistent with mitochondrial fusion being supported by increased protein levels. The process occurs as fast as 3 hours after proliferation begins. We elaborate below.

The mechanisms underlying mitochondrial dynamics remain incompletely understood (Salazar-Roa et al., 2017). However, emerging evidence indicates that mitochondrial fusion is regulated (at least in part) by protein levels. Specifically, it has been suggested that OPA1 is regulated by proteolysis and Mfn1 and Mfn2 by ubiquitin-mediated degradation (van der Bliek et al., 2013). Accordingly, we examined the protein levels of Mfn1, Mfn2, and OPA1 in our quiescent and proliferating fibroblasts. Indeed, we observed that mitochondria in proliferating cells have significantly higher levels of Mfn1, Mfn2, and OPA1 compared to mitochondria in quiescent cells. These data are now presented in Figure 2—figure supplement 2 of the revised manuscript. These data support that mitochondrial fusion involves increasing protein levels.

To assess how quickly mitochondrial fusion begins after cells start to proliferate, we needed to switch models. Our contact-inhibition system is well suited to study the transition from proliferation to quiescence upon contact inhibition at 100% confluence. It is not possible, however, to study the reverse transition from quiescence to proliferation by using contact inhibition. As such, to address the reviewers’ question, we established quiescence in 3T3 fibroblasts by serum starvation. We then induced proliferation by re-introducing serum and quantified mitochondrial elongation at three different time points (3 hours, 16 hours, and 30 hours). As expected, we observed that serum-starved quiescent cells have significantly shorter mitochondria. Notably, in as fast as 3 hours after serum was reintroduced to induce proliferation, there was a statistically significant increase in mitochondrial length. Mitochondrial length continued to increase with time. These data were added as Figure 2—figure supplement 5B.

2) One could argue that Mfn2 loss decreases OXPHOS overall rather than specifically preventing OXPHOS gain during cell proliferation. Can the authors show whether Mfn2 loss also decreases OXPHOS in quiescent cells?

We agree with this suggestion and performed the recommended experiment. Knocking down Mfn2 in quiescent cells led to only a minimal decrease in OXPHOS (~5%) compared to the substantial decrease in OXPHOS we observed upon knocking down Mfn2 in proliferating cells (~30%) (Author response image 1).

**Author response image 1. respfig1:** Comparison of Mfn2 knockdown on basal respiration in quiescent (**Q**) and proliferating (**P**) fibroblasts. Loss of Mfn2 had only minimal effects on oxygen consumption rate (OCR) in quiescent cells but had a substantial impact on proliferating cells. This plot is derived from data shown in Figure 2E and Figure 2—figure supplement 4, but is not included in the revised manuscript directly.

Since mitochondria in quiescent fibroblasts are already extensively fragmented (Figure 2D), we predicted that loss of Mfn2 would have little to no effect on OXPHOS. When we measured the oxygen consumption rates of scrambled siRNA controls (SSC) and Mfn2 knockdowns in the quiescent state, we found only a small difference. This contrasts to the substantial differences observed for proliferating cells (Figure 2E-F). These new data, which support that Mfn2 loss prevents gain of OXPHOS during proliferation rather than merely decreasing OXPHOS overall, are presented in Figure 2—figure supplement 4 of the revised manuscript.

3) Because the main readout of Mfn2 silencing is reduced proliferation (i.e. loss of fitness), the key experiments need to be controlled for off-target toxicity by re-expressing an siRNA-resistant cDNA of Mfn2.

As the reviewers suggested, we constructed a codon-modified cDNA that expresses wildtype Mfn2 but is resistant to the siRNA added (please see source file for sequences for siRNA and siRNA-resistant Mfn2). Expression of this siRNA-resistant Mfn2 (Mfn2^res^) in Mfn2 knockdowns restored Mfn2 protein level, mitochondrial respiration, and cellular proliferation. These data, which control for off-target toxicity, are presented in Figure 3—figure supplement 1 of the revised manuscript.

4) The M1 experiment is important because it would indicate that mitochondrial fusion is sufficient to drive proliferation. But more data are needed. Does this dose enhance fusion in these cells? Does it cause metabolic changes consistent with enhanced fusion? Is the effect on cell proliferation dose-dependent?

We agree with the reviewers that it is important to test whether increased mitochondrial fusion is sufficient to drive proliferation. In the original manuscript, we attempted to increase mitochondrial fusion pharmacologically with a drug known as M1. The reviewers suggested that we perform three M1-related experiments: (i) dose-response analysis for M1, (ii) quantify change in mitochondria length after M1 treatment, and (iii) analyze metabolic changes due to M1 treatment. We did these experiments but determined that M1 was not the best approach to test our predictions. Instead of using a drug to increase mitochondrial fusion, we found it to be more effective to overexpress Mfn2 in our wildtype 3T3 fibroblasts. As mentioned in response to point 1 above, previous studies have suggested that mitochondrial dynamics may be regulated by protein levels. Increasing levels of Mfn2 protein in fibroblasts increased mitochondrial respiration and cellular proliferation. Our results support that increased fusion is sufficient to drive proliferation.

First, we performed the three suggested experiments related to M1. Since we decided to remove M1 from the study (specifically Figure 3A in the original manuscript), we did not include these data in the revised manuscript.

We used EM imaging on DMSO controls and cells treated with M1. We did not observe a statistically significant difference between controls and M1-treated cells. Similarly, we did not observe a statistically significant difference in respiration between control cells and M1-treated cells.

Based on these data, M1 treatment did not induce enough mitochondrial fusion in our system to test whether an increase in mitochondrial fusion is sufficient to drive proliferation with the M1 drug. As an alternative strategy to test this prediction, we overexpressed wildtype Mfn2 (Mfn2^res^) in wildtype 3T3 fibroblasts. The overexpression resulted in a ~2-fold increase in protein level from whole-cell lysate (Figure 3—figure supplement 2A). The increase in protein led to a significant increase in mitochondrial respiration (Figure 3—figure supplement 2B) and a significant increase in proliferation rate (Figure 3—figure supplement 2C). These data are presented in Figure 3—figure supplement 2 of the revised manuscript.

5) Any insights into how HRAS induces PGC1α would improve the paper. The authors should also cite Weinberg et al. (2010) which first reported the role of oncogenic HRAS and KRAS on ROS and respiration.

We thank the reviewers for providing this reference, which we now cite. In terms of how HRAS induces PGC1α, the exact signaling cascade is not well defined (Dard et al., 2018). To the best of our knowledge, two pathways have been suggested as a potential link: the ERK/AMPK pathway (López-Cotarelo et al., 2015; Weinberg et al., 2010) and the KSR1 pathway (Fisher et al., 2011).

Based on these previous studies, we analyzed the expression levels and phosphorylation levels of ERK, AMPK, and KSR1. Phosphorylated AMPK and phosphorylated KSR1 were not detected. AMPK and KSR1 were not changed between empty vector controls (EV) and Ras transformed fibroblasts (Ras). We also did not observe any difference in Erk1/2 and phospho-Erk1/2 levels. These data are presented in Figure 4—figure supplement 2. They suggest that the mechanism for how Ras induces PGC1α in our fibroblast system may not involve a change in the ERK/AMPK or KSR1 pathways as has been suggested for other cells.

6) Are some of these metabolic changes enabled by changes in membrane transporter expression? Assessing the levels of SLC7A11 (cystine-driven glutamine anaplerosis) and SLC1A2/3 (glutamate/aspartate uptake) in the quiescent and proliferating states with and without Mfn2 expression might help explain some of these changes.

The metabolic changes that we observe do not seem to be enabled by changes in expression of the SLC7A11 or SLC1A2/3 membrane transporters between proliferating and quiescent cells. Consistent with this idea, however, we did find that cystine uptake was elevated by over 2-fold in wildtype proliferating cells compared to wildtype quiescent cells. In contrast, when Mfn2 was knocked down, cystine uptake was decreased by ~40%. Thus, while the observed metabolic changes may not be enabled by differential expression of membrane transporters, they do seem to be enabled by differential activity of the transporters.

We first analyzed the protein level of the cystine/glutamate transporter, SLC7A11, in quiescent and proliferating fibroblasts, with and without Mfn2 expression. We did not observe any differences in protein expression. In the revised manuscript, these data are shown in Figure 1—figure supplement 2A and Figure 3—figure supplement 3D. We combine the data here for convenience in Author response image 2.

**Author response image 2. respfig4:** Immunoblot of SLC7A11 from whole-cell lysates for scrambled siRNA control cells in the quiescent state (SSC_Q), siRNA control cells in the proliferating state (SSC_P), Mfn2 knockdown cells in the quiescent state (Mfn2^KD^_Q), and Mfn2 knockdown cells in the proliferating state (Mfn2^KD^_P).

To test whether the metabolic changes observed could be enabled by changes in transporter activity (independent of expression level), we evaluated the consumption rate of cystine. Strikingly, we found that cystine uptake was elevated by over two-fold in proliferating fibroblasts compared to quiescent fibroblasts. In contrast, when Mfn2 was knocked down, cystine uptake decreased by ~40%. These data were added as Figure 1—figure supplement 2 and Figure 3—figure supplement 3D.

As the reviewers suggested, we also assessed the protein levels of the glutamate/aspartate transporter SLC1A2/3 (Excitatory Amino Acid Transporter, EAAT2/1). In addition, we assessed the protein level of another glutamate/aspartate transporter SLC1A1 (EAAT3). Proliferation did not affect the protein expression of any of these transporters, while Mfn2 knockdown seemed to decrease the expression of SLC1A3 but not SLC1A1/2 (see Author response image 3). Since these transporters can carry amino acids in either direction (i.e., in or out of a cell), it is difficult to interpret whether this change in transporter expression is the cause of the change in uptake or excretion rates observed in Figure 3B,3E.

**Author response image 3. respfig5:** Immunoblot of SLC1A1/2/3 transporters from whole-cell lysates for scrambled siRNA control cells in the quiescent state (SSC_Q), siRNA control cells in the proliferating state (SSC_P), Mfn2 knockdown cells in the quiescent state (Mfn2^KD^_Q), and Mfn2 knockdown cells in the proliferating state (Mfn2^KD^_P).